# Assessment of Soybean Lodging Using UAV Imagery and Machine Learning

**DOI:** 10.3390/plants12162893

**Published:** 2023-08-08

**Authors:** Shagor Sarkar, Jing Zhou, Andrew Scaboo, Jianfeng Zhou, Noel Aloysius, Teng Teeh Lim

**Affiliations:** 1Division of Plant Science and Technology, University of Missouri, Columbia, MO 65211, USA; ssw5r@umsystem.edu (S.S.);; 2Department of Biological Systems Engineering, University of Wisconsin-Madison, Madison, WI 53705, USA; 3Department of Chemical & Biomedical Engineering, University of Missouri, Columbia, MO 65211, USA

**Keywords:** crop breeding, high-throughput phenotyping, remote sensing, image features

## Abstract

Plant lodging is one of the most essential phenotypes for soybean breeding programs. Soybean lodging is conventionally evaluated visually by breeders, which is time-consuming and subject to human errors. This study aimed to investigate the potential of unmanned aerial vehicle (UAV)-based imagery and machine learning in assessing the lodging conditions of soybean breeding lines. A UAV imaging system equipped with an RGB (red-green-blue) camera was used to collect the imagery data of 1266 four-row plots in a soybean breeding field at the reproductive stage. Soybean lodging scores were visually assessed by experienced breeders, and the scores were grouped into four classes, i.e., non-lodging, moderate lodging, high lodging, and severe lodging. UAV images were stitched to build orthomosaics, and soybean plots were segmented using a grid method. Twelve image features were extracted from the collected images to assess the lodging scores of each breeding line. Four models, i.e., extreme gradient boosting (XGBoost), random forest (RF), K-nearest neighbor (KNN) and artificial neural network (ANN), were evaluated to classify soybean lodging classes. Five data preprocessing methods were used to treat the imbalanced dataset to improve classification accuracy. Results indicate that the preprocessing method SMOTE-ENN consistently performs well for all four (XGBoost, RF, KNN, and ANN) classifiers, achieving the highest overall accuracy (OA), lowest misclassification, higher F1-score, and higher Kappa coefficient. This suggests that Synthetic Minority Oversampling-Edited Nearest Neighbor (SMOTE-ENN) may be a good preprocessing method for using unbalanced datasets and the classification task. Furthermore, an overall accuracy of 96% was obtained using the SMOTE-ENN dataset and ANN classifier. The study indicated that an imagery-based classification model could be implemented in a breeding program to differentiate soybean lodging phenotype and classify lodging scores effectively.

## 1. Introduction

Soybean is one of the major grain productions in the USA, and a total of 87.6 million acres of land were planted in 2021 with an average soybean yield of 3.45 metric tons per hectare, according to the report of the United States Department of Agriculture (USDA), National Agriculture Statistics Service [1]. However, soybean yield may be significantly affected by environmental conditions [2]. Breeding programs aim to develop new crop varieties with improved traits, such as improved yield, resistance to biotic and abiotic stresses, desired maturity stage and no lodging. Lodging occurs when the plant stems break or bend over, and the plants displace entirely from their original positions. Lodging is a morphological trait, and lodged soybean plants can significantly reduce yield. Lodging in different growing stages can cause different yield reductions, such as by 18–32% at the R5 stage [3], 12–18% at R3, and 13–15% at stage R6 [4]. Furthermore, complete lodging at the full maturity stage can cause more than a 30% yield reduction [5]. Therefore, developing soybean cultivars with lodging tolerance is essential to enhance productivity and improve yield stability [6].

Improvement of lodging resistance cultivars not only improves yield by increased solar radiation interception but also improves mechanical harvesting efficiency [7]. Lodging tolerant soybean cultivars selection is one of soybean breeders’ efforts in developing high-yielding soybean cultivars. Soybean lodging can be caused by environmental factors, such as moist and fertile soil, densely populated lines, soybean stem borer [8], and meteorological conditions [9]. Lodging can also be caused by external forces, wind, rain, hail [10] and morphological parameters of the cultivars. Numerous studies have been conducted on rice, maize, oats, barley, and canola lodging to determine the impact on crop yield and development [11,12,13,14]. It is essential to develop lodging-resistant cultivars. Conventional breeding programs quantify lodging tolerance traits based on visual observation and score them from 1 to 5. Visual observation and scoring are subjected to human errors and biases that may hinder the breeding efficiency and to identify the right genotypes [15,16]. Meanwhile, manual measurement and visual observation are time-consuming, laborious, and costly [17]. Therefore, developing a high throughput selection approach for quantifying soybean lodging scores using remote sensing technologies is a prime concern.

Remote sensing systems used in agriculture, including precision agriculture (PA), can be classified based on the sensor platform and the type of sensor employed. These platforms typically include satellites, aerial platforms such as aircraft and unmanned aerial vehicles (UAVs), and ground-based systems [18]. The sensors used for remote sensing vary in their spatial, spectral, radiometric, and temporal resolution capabilities [19]. Spatial resolution refers to the size of the pixel representing an area on the ground, with high-resolution sensors having small footprints and low-resolution sensors having larger footprints. Temporal resolution is related to the sensor platform’s orbit time, determining how frequently it revisits the same observation area. Spectral resolution, on the other hand, is indicated by the number of bands captured in the electromagnetic spectrum range [20]. Remote sensing systems have been used to quantify crucial factors of plants, such as photosynthesis, plant nutrients, and various biotic and abiotic stresses (e.g., disease and water stress) [21,22,23]. While many modern satellites offer high spatial resolution (less than 5 m) and frequent temporal coverage (daily), publicly available satellite products may still be too coarse for various precision agriculture applications. The optimal spatial-temporal resolution needed for PA depends on management goals, field size, and the equipment’s capability to adjust input application rates (e.g., irrigation, fertilizer, pesticide) [18].

Unmanned aerial vehicle (UAV) based imagery sensors such as digital and multispectral cameras have been used previously to select breeding lines using image-derived features of breeding rows within abbreviated time [24]. UAV image-derived features have been used by numerous researchers to quantify crop responses to biotic and abiotic stress [25], such as drought stress [26], salt [27], and pest resistance [22]. UAV-based remote sensing techniques have been used for non-crops as well. For example, turfgrass drought phenotyping was studied by Yousfi [26] to select drought-tolerant genotypes for sustainable green spaces in water-limited regions. Evaluating six turfgrass mixtures under different water regimes, vegetation and water status indices showed significant differences in growth. Normalized Difference Vegetation Index (NDVI) and Green Area (GA) distinguished drought-tolerant and susceptible species. At the same time, drone-derived GA proved more helpful in assessing turfgrass variability than ground RGB images or NDVI. Pest-resistant genotypes of potatoes have been studied based on the estimation technique for disease severity in a field using RGB imagery from UAV [28]. Furthermore, image features show an active correlation with soybean traits such as maturity, plant height, yield, flowering time, wilting and canopy size [15,29,30,31,32,33], which are particularly important in breeding programs.

Studies have been conducted to identify lodging-resistant genotypes of different crops in general and for breeding purposes, such as rice, maize, oats, barley, and canola [11,13,14,16] using remote sensing technologies. Image features have been used for lodging detection, such as textural features and Gray-Level Co-occurrence Matrix (GLCM), were used for wheat and canola lodging detection [34,35]. In addition, Gabor filtering was used by Mausda [36] to detect the degree and direction of rice lodging. [37,38] investigated and found that canopy spectral reflectance increased and NDVI decreased with the high severity of lodging. [39] also used textural features adopting an airborne to improve wheat lodging classification accuracy. [24] combines spectral features with textural features for the rice lodging classification.

Many machine learning and deep learning models have been used to identify crop lodging. By incorporating the digital surface model (DSM) and texture information alongside spectral data, the decision tree classification model achieved an impressive accuracy of 96.17% and successfully identified rice lodging ratios [40]. A comprehensive methodology for lodging identification in maize using UAV imagery and logistic regression effectively predicts the occurrence probability of lodging. It reveals the importance of canopy structural features in discriminating lodging at the plot scale [41]. Nearest neighbors, linear discriminant analysis, random forest (RF), neural network (NN), and support vector machine (SVM) are extensively used to distinguish crops. Rajapaksa et al. [34] utilized texture features and applied SVM to distinguish between wheat and canola lodging.

Similarly, Liu et al. [42] employed SVM on color, texture, and thermal features to differentiate between lodging and non-lodging in crops. Deep learning, a modern machine learning approach associated with artificial intelligence for image processing and data analysis [43], has recently gained prominence. The convolutional neural network (CNN) within deep learning can simultaneously function as a feature extractor and classifier.

However, within a breeding field, the number of lodged plots is much lower than that of non-lodged plots under natural conditions, which imbalanced the dataset in the analysis process. Imbalanced data may cause increased errors and decreased robustness of developed models. To improve the accuracy, imbalanced datasets are usually preprocessed using Synthetic Minority Oversampling Technique (SMOTE) and the Edited Nearest Neighbors under-sampling algorithm (SMOTE-ENN) [44].

Though research has been conducted using machine learning and deep learning methods to detect crop lodging, to the best of our knowledge, there is no research conducted on crop lodging detection for soybean breeding using the UAV imagery technique. Hence, this study aimed to investigate the potential of quantifying lodging scores of soybean breeding lines using UAV-based imagery and machine learning methods. The textural image features provide Appendix A about the object properties, which can help the heterogeneous crop field assessment [45]. The specific objectives were: (1) to develop an RGB image texture feature-based soybean lodging classification model using machine learning algorithms and (2) to assess the classification accuracies among different classification models. The current research makes the following contributions:Imbalanced dataset processing: This study has contributed by outlining the methods used for texture feature extraction, preprocessing of imbalanced datasets, including image feature extraction, feature selection, and applying various resampling techniques such as SMOTE-Tomek Link, Synthetic Minority Oversampling-Edited Nearest Neighbor (SMOTE-ENN), Borderline-SMOTE, Synthetic Minority Oversampling-Nominal Continuous (SMOTE-NC), and Adaptive Synthetic (ADASYN) to address the class imbalance issue in the lodging dataset.Efficient lodging assessment for soybean breeding: For the first time, evaluated the potential and methods to assess the lodging score of soybean breeding lines in field conditions using UAV-based imagery and machine learning.

## 2. Results

### 2.1. Feature Selection

Twelve texture features were selected using the random forest recursive feature elimination (RF-RFE) method with two others (Entropy and Information measures of correlation I) eliminated from all the features. The result shows that the 12 features were essential to obtain the highest score by performing the RFE algorithm with cross-validation. Figure 1 shows that the RFE uses a random forest algorithm to test the combination of features, where the number of features was 12 with a maximum score of 0.754. Yellow line in the Figure 1 indicates the feature scores obtained during the RF-RFE process. The highest point on the yellow line indicates the maximum score achieved during the RF-RFE process. This point corresponds to the set of features that leads to the best performance for predicting soybean lodging. These twelve features, one target, and 1266 observations (plots) were used to analyze the soybean lodging using four machine learning classifiers.

### 2.2. Original and Balanced Dataset

Table 1 shows the number of data points included in each class for the original dataset and four classifiers. Each classifier has five preprocessing methods datasets. The original dataset was highly imbalanced, where the non-lodging class was 76%, medium lodging was 16%, high lodging was 6%, and severe lodging was 0.08% of all datasets. To overcome the imbalance issue, five preprocessing (SMOTE-Tomek Link, SMOTE-ENN, Borderline-SMOTE, SMOTE-NC and ADASYN) methods were used to balance the dataset, and the number of data points included in each preprocessing method for each class are shown in Table 1.

### 2.3. Classification Performance of Machine Learning Models

#### Classification Performance of Four Machine Learning Models Using Original and SMOTE-ENN Dataset

Using the original dataset for the classification of soybean lodging, we found that XGBoost, Rf, KNN, and ANN showed an overall accuracy of 0.80, 079, 0.77, and 0.73, respectively. All the machine learning classifiers only showed higher precision, recall, and F1-score for the NL class. No model correctly classified other classes of ML, HL, and SL. Because there were very few numbers of data points associated with ML, HL, and SL classes. More precisely, 36 data points in ML, 12 data points in HL, and only 4 data points were associated in the testing set for the classification using the original dataset. Due to this limited number of data points, classifiers couldn’t recognize any of them, resulting in poor precision, recall, and F1-score for these classes. The confusion matrix and model performance metrics of four lodging classes using the original dataset are shown in Appendix A. Overall accuracy, kappa value, and misclassification rate are shown in Appendix A and due to the poor results using the original and imbalanced dataset, used five data balancing methods (SMOTE-Tomek Links, SMOTE-ENN, Borderline-SMOTE, SMOTE-NC, and ADASYN). The following section describes the results of ML classifiers using five balancing methods.

After using five balanced datasets, the study found that the SMOTE-ENN dataset provided the best accuracy, higher precision, recall, F1-score, kappa coefficient and lower misclassification rate than the other four datasets. Table 2 represents the evaluation results of XGBoost, RF, KNN and ANN classifiers using the SMOTE-ENN balanced dataset for classifying four soybean lodging classes (NL, ML, HL and SL). Evaluation results using other balanced datasets are provided in the Appendix A.

The results of the XGBoost model shown in the table show higher classification accuracy of 0.94 (Figure 2) using the SMOTE-ENN dataset, whereas, using other datasets, the classification accuracy ranges between 0.84–0.89 (Appendix A). SMOTE-ENN dataset outperformed other datasets. Higher precision, recall and F1-score were obtained for SL classes with the values of 0.99, 0.99 and 0.99, respectively. On the other hand, a higher kappa value of 0.91 was also obtained using the SMOTE-ENN dataset. The overall accuracy, kappa coefficient and misclassification rate of each class for the other four (SMOTE-Tomek Links, Borderline-SMOTE, SMOTE-NC, and ADASYN) balanced datasets are shown in Appendix A. Among all other datasets, a lower misclassification rate of 12% was obtained using the SMOTE-ENN dataset. A comprehensive comparison showed that using SMOTE-ENN was the most ideal dataset, with the higher area under the receiver operating curve (AUC) from the ROC curve and the average precision (AP) values from the precision-recall curve Figure 3a and Figure 4a. Furthermore, the RF classifier resulted in higher overall classification accuracy of 0.93 was obtained using the SMOTE-ENN dataset, whereas 0.85, 0.88, 0.86 and 0.85 (Appendix A) were obtained from SMOTE-Tomek Link, Borderline-SMOTE, SMOTE-NC and ADASYN, respectively. SMOTE-ENN outperformed all other methods with higher precision, recall, F1-score, and kappa value. The misclassification rate was also minimum for the SMOTE-ENN dataset with a value of 13%. Overall accuracy, kappa coefficient and misclassification rate of four balanced datasets are shown in figure (Appendix A). The SMOTE-ENN dataset showed better results than other datasets, resulting in the best AUC from ROC curve AP values from the precision-recall curve Figure 3b and Figure 4b.

Unlike other classifiers XGBoost and RF, KNN showed a similar trend in classifying soybean lodging. Higher overall classification accuracy was obtained using the SMOTE-ENN dataset with a value of 0.91 (Figure 2), where NL class showed lower recall and f1-score of 0.55 and 0.69, respectively. Other classes, ML, HL and SL, showed consistency with a value greater than 0.75 in all respect. A higher kappa value of 0.87 and a lower misclassification rate were also obtained from SMOTE-ENN, with a value of 17% (Figure 2). Other results (Appendix A and Appendix A) using a balanced dataset aren’t as good as SMOTE-ENN. The AUC values for each class were 0.96, 0.98, 1.00, and 1.00 for NL, ML, HL, and SL, respectively, whereas AP values were 0.90, 0.96, 0.99 and 1.00 (Figure 3c and Figure 4c) for SMOTE-ENN dataset. As per previous results, SMOTE-ENN showed a higher classification accuracy of 0.96 (Figure 2) with higher precision, recall and f1-score for ML, HL and SL, except for the class NL (Table 2**).** A very minimum misclassification rate of 7% was obtained using the SMOTE-ENN dataset, whereas other datasets resulted in a 32%, 25%, 25% and 36% misclassification rate (Appendix A). Other results using SMOTE-Tomek Link, Borderline-SMOTE, SMOTE-NC and ADASYN datasets showed classification accuracy higher than 0.8, but SMOTE-ENN outperformed in every case. The ROC curve using four (SMOTE-Tomek Links, Borderline-SMOTE, SMOTE-NC, and ADASYN) balanced datasets along with XGBoost, RF, KNN, and ANN classifiers are shown in Appendix A, respectively. Furthermore, the precision-recall curve is shown using four (SMOTE-Tomek Links, Borderline-SMOTE, SMOTE-NC, and ADASYN) balanced datasets along with XGBoost, RF, KNN, and ANN classifiers are shown in Appendix A, respectively. On the other hand, higher AUC and AP values were obtained from the SMOTE-ENN dataset for each class Figure 3d and Figure 4d.

## 3. Discussion

Lodging is one of the major factors of reduced crop yields [46]. Accurate classification of crop lodging is highly economical for decision-making regarding breeding purposes. Some researchers already used UAV RGB imagery to classify the lodging of different crops with the increasing development of UAV technology. But to our knowledge, there is no research conducted to classify soybean lodging for breeding purposes. One of the reasons might be the complexity of analysis using an imbalanced dataset and focusing on mostly known phenotypic traits leads to putting less focus on soybean lodging classification using UAV-based imagery. Though there was a big challenge associated with the data preprocessing as the lodging classes were highly imbalanced, it came to a point to justify the results with other researchers’ findings. The original dataset and five data balancing methods (SMOTE-Tomek Link, SMOTE-ENN, Borderline-SMOTE, SMOTE-NC and ADASYN) were used to balance the dataset. They analyzed these balanced datasets using XGBOOST, RF, KNN and ANN classifiers. From the above-described results, SMOTE-ENN performed the best among all balancing techniques for all the machine learning classifiers, achieving the highest overall accuracy, lowest misclassification rate and highest Kappa coefficient in all the cases. Based on the SMOTE-ENN dataset results of four classifiers, the comparison was described in the following section.

Table 3 shows the comparative analysis of crop classification results among soybean lodging classification and other crop lodging classifications done by different researchers. RGB image features for maize lodging severity with an accuracy of 94.5% with the XGBoost classifier after using SMOTE-ENN preprocessing [47]. The researchers used before and after balancing the dataset method for maize lodging classification, where the non-lodging and lodging data distribution was 713 and 87 before balancing, respectively. After balancing using the SMOTE-ENN method, the non-lodging and lodging dataset becomes 546 and 671, respectively, which reduces the non-lodging dataset and increases the lodging dataset for balancing the dataset [47]. Undersampling and SMOTE method for wheat yield estimation by Chemchem [48], where the researcher finds out that the SMOTE data balancing method significantly increases training score and AUC ROC values using well-known machine learning methods. Random forest classifier showed the training score and AUC ROC values of 99.40% and 0.72 (without sampling), 99.78% and 0.73 (undersampling), and 99.92% and 0.90 (SMOTE sampling), respectively. Hyperspectral imagery data available from the public domain with 16 classes were classified using tree-based ensembled classifiers by Datta [49], encompassing several data balancing methods, which resulted in SMOTE, Tomek-Links and their combinations of higher accuracy and best resampling strategy.

This study followed the same data balancing method, SMOTE-ENN, where the number of NL datasets (Table 1) was reduced, and the number of ML, HL, and SL increased using the data balancing method. Our study resulted in 94% of soybean lodging classification accuracy with the XGBoost classifier and SMOTE-ENN data balancing method. RGB image features were used for grass lodging severity and found an accuracy of 79.1% [50], whereas 86.61% accuracy was observed by [12] for maize lodging classification using the maximum likelihood classification (MLC) algorithm. On the other hand, a random forest (RF) classifier was used for barley lodging mapping and found an overall accuracy of 99.7% using multispectral imagery [14]. This study found an overall accuracy of 93% on the testing dataset using an RF classifier for soybean lodging classification. The superiority of other researchers’ results using RF may be associated with its data feature extraction and the number of training and testing balanced datasets. Furthermore, KNN yields classification accuracy of 91% for soybean lodging, whereas nearest neighborhood classification and support vector machine (SVM) by [34,38] reported wheat lodging classification at 90% and 92.6%, respectively, using multispectral imagery. Furthermore, Principal component analysis and ANN was used to differentiate the rice lodging with an overall accuracy of 97.8% using hyperspectral reflectance data [51], whereas our study reveals an overall accuracy of 96% using RGB image-derived textural features for soybean lodging. The higher accuracy obtained by [52] might be the reason for using the hyperspectral imagery technique, which can identify and quantify molecular absorption. Soybean lodging, which can decrease crop yield and quality, can be minimized by accurately classifying soybean lodging. Breeders can identify and address the factors associated with lodging, such as genetics, environmental conditions, or other agronomic factors. On the other hand, soybean lodging classification based on a deep learning algorithm can automatically extract intrinsic features from the dataset using different supervised and unsupervised learning to classify the different extents of soybean lodging. The best advantage of the RGB imagery technique is low cost and beneficial for smallholders to detect crop lodging.

### Future Work

There is still a huge lack of soybean lodging detection research for breeding. Though there are other traits such as yield, plant height, days to maturity, and leaf wilting are associated with selecting the best genotypes for breeding purposes, lodging is one of the crucial factors considered by the breeders, which is somehow not in mainstream research using UAV based imagery technology. One of the biggest limitations in soybean lodging detection is the imbalanced dataset of different classes, which challenges researchers in analyzing and obtaining good output. To address the emerging issues, four balancing methods were used to balance the class difference. Further, detailed classification exploring other supervised deep learning and machine learning methods can be explored to obtain higher accuracy for soybean lodging detection.

## 4. Materials and Methods

### 4.1. Field Experiment

The field experiment was conducted in 2019 at the Bay Farm Research Facility (38°54′08.3″ N, 92°12′29.8″ W), Columbia, Missouri, United States. The field is in a humid subtropical region (Köppen climate classification code: Cfa) [53]. The experimental field was a preliminary yield trial (PYT) to select advanced yield trial (AYT) genotypes. The breeding lines were observed for several traits throughout the growing and harvesting period. One thousand and seven hundred seventy-three (1773) soybean genotypes were planted in four-row plots with a row length of 3.6 m and row spacing of 0.8 m on 3 June 2019 (without replicates). The distance between each plot was the same as the row spacing, and the end-to-end spacing between the two plots was 0.6 m. Weather conditions during the growing season showed in Figure 5, which includes daily average temperature and cumulative precipitation. The weather data was acquired from the nearby weather station (Bradford Research and Extension Center), a part of Missouri Mesonet—Weather Station Network, located within 1.01 (km) of the studied field plots.

### 4.2. Ground Data Collection

Lodging was rated on a 1 to 5 scale with 0.5 increments by the experienced breeders at the maturity stage R8. A rating of 1 showed that all plants and branches were erect, 2 showed that all plants were leaning slightly or a few plants down, 3 demonstrated that plants were leaning moderately (45 degrees), or 25% to 50% of the plants down, 4 showed that all plants leaning considerably, or 50% to 80% of the plants down and 5 showed that all plants are down. These ratings are general requirements that are followed by the breeders to observe visually and to score each plot. All plots were then assigned into four classes based on their scores, namely no lodging (NL, score 1.0, 1.5), medium lodging (ML, score 2.0, 2.5), high lodging (HL, score 3.0, 3.5) and severe lodging (SL, scores 4.0, 4.5 & 5.0). The assigned four classes based on the lodging score were developed for this study after discussing with the experienced soybean breeders. Some examples of plots with different lodging scores are shown in Figure 6. A total of 1773 plots were processed, among which 1266 plots were used for the image feature calculation, and the rest of the plots were filler and discarded. In this study, the number of plots of each lodging class NL, ML, HL, and SL is non-lodging, medium lodging, high lodging, and severe lodging, respectively.

### 4.3. UAV Imagery Data Collection

The aerial images were collected at the full seed development stage of R6 on 29 August 2019 using a UAV platform (DJI Matrice 600 Pro, DJI, Shenzhen, China) equipped with a Sony A6300 (Sony Corporation, Tokyo, Japan) camera. Images were taken at 0.5 frames per second (fps) with a resolution (number of pixels) of 6000 × 4000 pixels at a flight height of 30 m above the ground level with an overlap of 80% for both sides of the image. During the image acquisition, the camera was set to shutter 1/1000 s, ISO 100–200, F-stop auto and daylight mode. A Real-Time Kinematic (RTK) GNSS positioning system (Reach RS+, Emlid, St. Petersburg, Russia) was used to obtain the GNSS coordinates of the GCPs. To ensure sufficient satellite reception, the base station was mounted on a tribrach fixed in an open field area. The base position was obtained in the initial setting by accumulating its GNSS coordinates for 30 min. The base station was placed at the same location as each data collection. The rover receiver was placed vertically in the holes on a monopod after the GCPs were pulled out, and the GNSS coordinate of each GCP was recorded after accumulating for 10 s using ReachView (Emlid, St. Petersburg, Russia). The geo-referencing information of each image was recorded in a separate.csv file. Images were stitched using Agisoft PhotoScan Pro (v1.3.4, St. Petersburg, Russia) for further processing.

### 4.4. Image Processing

Figure 7 illustrates this study’s image processing and data analysis pipelines. First, the texture feature was extracted from RGB orthomosaic after a few preprocessing steps. The optimal features were selected using Random Forest-Recursive Feature Elimination (RF-RFE) method. Five resampling methods, namely Synthetic Minority Oversampling (SMOTE)-Tomek Links, SMOTE-Edited Nearest Neighbor (ENN), Adaptive Synthetic Oversampling (ADASYN), Borderline-SMOTE and SMOTE-Nominal Continuous (NC) method were compared for treating the data imbalance. At last, the performance of four machine learning models in the lodging classification was compared.

An orthomosaic image was generated using Agisoft PhotoScan Pro. Following the procedures described by [27]. The orthomosaic image was exported as .*tif* image and was processed using the Matlab Image Processing Toolbox and Computer Vision System Toolbox (ver. 2021b, The MathWorks, Natick, MA, USA). Individual soybean plots were separated from orthomosaic images by manually cropping a rectangle region of interest (ROI) around each plot. The size of each plot varied due to the planting error. Hence the size of ROI also varied to cover each soybean plot according to its width and length. The background of each image (e.g., soil, leaf shadow and plant residue) was removed by detecting the projected canopy contours using the “activecontour” function [54] with the “Chan-Vese” method [55]. The foreground consisting of soybean plants is the pixels within a full contour, and the outside of the contours is the background. Contours with extremely small regions were detected as noises using the “regionprops” function and removed from the foreground.

### 4.5. Texture Feature Extraction and Selection

A variety of image features could be used in machine learning to classify lodging and non-lodging plots accurately. Color image features that show variation in the leaves or stem color of the plant and don’t represent the lodging or non-lodging condition of the plots. On the other hand, lodging plots most likely have non-uniform and heterogenous patterns and non-lodging plots have uniform and homogenous patterns [43]. Considering these factors, textural features were extracted from the selected plots. The texture is one of the essential features of digital image processing. Numerous researchers have vastly used it for processing remote sensing data such as object detection, image classification, crop trait classification, image evaluations, and lodging detection [35,56,57,58].

Furthermore, to reduce the noise in isolated pixels in classification, texture information is one of the prime features for classification [58]. Gray Level Co-occurrence Matrix (GLCM) was first developed by [59], and the GLCM considers pairs of pixels separated by a certain distance and oriented at a specific angle within the image. It calculates the frequency of occurrence of these pixel pairs with different combinations of grey levels [60]. To calculate the texture features, the original image is first converted to a grayscale image. Then the features of the grayscale images are extracted using the relationship of brightness values between the center pixel and its neighborhood pixel within the predefined kernel. The GLCM can produce different texture information according to its grayscale level, kernel size and direction by using the relationship. The selected texture features used in this study are listed in Table 4. Those features were angular second moment, contrast, correlation, variance, inverse difference moment, sum average, sum variance, sum entropy, entropy, difference variance, difference entropy, information measures of correlation I, information measures of correlation II, and maximum correlation coefficient.

The feature selection step eliminates the less important feature at each iteration. The random forest recursive feature elimination (RF-RFE) method [72] was used to select the optimal texture features. This recursive process ranks features according to the feature importance given by the RF. The texture feature set was continuously reduced through iterative loops to select the required features. The 10-fold cross-validation strategy was used to select the best basis function for the feature selection.

### 4.6. Preprocess Imbalanced Data

In this study, the sample number of each lodging class was imbalanced, with only a few high and severe lodging plots. Data balancing or resampling are commonly used methods to tackle the class imbalance issue in machine learning models by isolating it from the classification algorithms. Five preprocessing methods (i.e., SMOTE-Tomek Links, SMOTE-ENN, Borderline-SMOTE, SMOTE-NC and ADASYN) were compared to identify the suitable models for the imbalanced data in this study.

Synthetic Minority Oversampling Technique (SMOTE) algorithm is a resampling method based on a random oversampling algorithm that generates synthetic samples by the difference between adjacent minority samples [73]. The process includes the selection of an example of *x* minority class and the nearest minority class neighbors. The synthetic data is created by choosing one of the *k* nearest neighbors *y* at random, then connecting x and y to form a line segment in space characteristics. Then the synthetic dataset is produced as a combination of two selected samples x and y [74]. SMOTE can increase the number of minority classes in the dataset [75,76]. Several oversampling methods have been derived using the SMOTE as a basis [77], which includes SMOTE-Tomek Links, SMOTE-ENN, Borderline SMOTE, and SMOTE-NC in this study.

The SMOTE-Tomek Links method involves the integration of SMOTE oversampling and Tomek Links undersampling techniques [78]. It generates synthetic data for the minority class using SMOTE and removes data identified as Tomek Links from the majority class. The SMOTE-ENN combines SMOTE and Edited Nearest Neighbor (ENN), an undersampling method. SMOTE-ENN enhances the accuracy of classifying minority classes by eliminating observations from the majority classes near the class boundary of distinct classes calculated through the nearest neighbor algorithm [79]. One of the main characteristics of the SMOTE-ENN method is that the processed data does not have the same number of instances in different classes. Instead of having equal instances in each category, the resampled majority class will still be larger than the minority class, but the size difference between the categories will be reduced. In addition, Borderline-SMOTE is another technique used for oversampling in datasets that are not balanced. The Borderline SMOTE technique enhances the distribution of samples within a dataset by generating new samples based on a small number of samples located at the boundary [80]. Lastly, SMOTE-NC is an extension of SMOTE that considers several factors, such as the nearest neighbors of the minority instances, the median of the standard deviation of both nominal and continuous variables, the Euclidean distance between the minority instance and its k-nearest neighbors, and the desired ratio of the minority class over the majority class [81,82].

On the other hand, this study also tested the Adaptive Synthetic (ADASYN) sampling approach. Similar to SMOTE method, ADASYN generates more synthetic samples for the minority classes along the linear function by weighting distance [83] and according to the level of difficulty in learning [74]. ADASYN utilizes a weighted distribution for diverse minority classes to determine the number of artificial samples produced for each minority category [84].

### 4.7. Machine Learning Models for Soybean Lodging Classification

The extreme gradient boosting (XGBoost) method is y for classification. XGBoost controls the overfitting by using the regularized model formalization, which resulted in better performance compared to the previous boosted algorithms [85]. XGBoost consists of a few hyperparameters such as *nrounds* (the number of trees), *eta*, *learning rate* and *depth* (the depth of the tree) [86] that can be optimized to improve the performance. To optimize the XGBoost model performance, a nested cross-validation approach was applied to find the optimal hyperparameter factors to produce the best models. Hyper-parameter was tuned for each preprocessed dataset and included in the Appendix A. The study uses Tree-Structured Parzen Estimator (TPE) method to tune the hyper-parameters, which involves defining the hyper-parameter spaces and distributions. The aim was to develop a high-precision and recall model for the soybean lodging classification.

An ensemble classification algorithm, a random forest (RF) model, was used to classify soybean lodging based on image features. RF is a group of tree-based classifiers and uses bootstrapping to improve the diversification of classification trees. Random Forest takes advantage of the high speed and accuracy of the decision tree algorithm for classification problems by generating multiple decision tree models. Each decision tree is independent, and the errors are minimized collaboratively, resulting in more accurate and reliable classification results [87]. The grid search method was used with 5-fold cross-validation to optimize the RF model performance and find optimal hyper-parameters. Six hyper-parameters were tuned using the grid search method for each treated dataset, including the original dataset. The best values of hyper-parameters are included in the Appendix A). RF was used to develop a soybean lodging classification model using these best hyper-parameter values.

The K-nearest neighbor (KNN) is a non-parametric supervised machine learning algorithm, one of the popular algorithms for data processing and modeling [88]. The KNN algorithm was used to classify the soybean lodging genotypes selection, where the primary parameter “n_neighbor” = 3 was set for the analysis.

Artificial neural networks (ANNs) have been extensively used to classify crops and crop traits [89,90,91]. ANN contains an input layer, multiple hidden layers, and one output layer. Each hidden layer contains various numbers of neurons and quantifies the related mathematical equations to identify the complex relationship between the data and the input layer and the data in the output layer. An artificial neural network can be constructed by tuning two hyperparameters, such as the number of nodes in the hidden layer and the number of control iterations [52].

### 4.8. Data Analysis and Accuracy Assessment

All the analysis and modeling were performed in Google Colaboratory (Colab) Pro. Machine learning classifiers (XGBoost, RF, KNN and ANN) were compared regarding their performance in classifying the four lodging classes. The balanced dataset was split into training and testing, with 80% and 20%, respectively. The model performance was evaluated using a 5-fold cross-validation (CV) to calculate the classification accuracy.

The four classes of soybean lodging scores were assessed based on the number of samples that were correctly or falsely classified as either presence (True Positive, TP or False Positive, FP) or absence (True Negative, TN or False Negative, FN) using XGBoost, RF, KNN, and ANN models. A group of metrics were used for evaluation and calculated using Equations (1)–(5). The evaluation was based on several performance metrics: precision, recall, F1-score, kappa, and overall accuracy (OA). The precision indicates the proportion of correctly predicted presences, recall represents the ratio of correctly predicted positive samples, and F1-score is the harmonic mean of precision and recall. The F1 score measures model accuracy that balances precision and recall, which ranges from 0 to 1, with 1 being the best possible score, indicating perfect precision and recall. The Kappa value represents the proportion of correctly predicted sites, and OA indicates the overall accuracy of the model.
(1)$$Overall\;accuracy\;(OA)=\frac{No.\;of\;samples\;classified\;correctly\;in\;a\;test\;set}{Total\;No.\;of\;samples\;in\;a\;test\;set}\times 100\%$$
(2)$$Kappa=\frac{Po-Pe}{1-Pe}$$
where, P_0_ = is the overall accuracy of the model; P_e_ = is the measure of the agreement between the model predictions and actual class values
(3)$$Precision=\frac{TP}{TP+FP}$$
(4)$$Recall=\frac{TP}{TP+FN}$$
(5)$$F1\;score=\frac{2\times (Precision\times Recall)}{\left(Precision+Recall\right)}$$

## 5. Conclusions

In this study, the research aimed to classify the soybean lodging extent using UAV RGB imagery technology for breeding purposes. The RGB image-derived textural features and RGB image were tested to classify the different soybean lodging extent. The images were preprocessed by adjusting the grids according to each plot shape, cropping and background removal to generate the texture features for machine learning models. The experimental results indicate that the developed classification models precisely distinguished high lodged and severe lodged soybean genotypes with different genotype backgrounds with higher precision and recall using every machine learning deep learning model.

The classification performance of the machine learning models using RGB image-derived texture features was 94%, 93%, 91% and 96% using SMOTE-ENN XGBoost, SMOTE-ENN RF, SMOTE-ENN KNN and SMOTE-ENN ANN, respectively. In addition, image features and overall classification accuracies of different machine learning models show consistency and a similar pattern in classifying four classes which can be more justified using different environments datasets of non-tested genotypes.

The findings of this study underscore its significant potential impact on soybean breeding programs and agricultural practices. By utilizing UAV-based RGB imagery technology and machine learning methods, this research presents a precise and efficient approach to classify soybean lodging extent, providing breeders with a valuable tool for improving crop resilience and productivity. The ability to accurately distinguish different soybean lodging levels using machine learning models enhances breeders’ capacity to identify superior genotypes and select lodging-resistant varieties. The high classification performance achieved by various machine learning models demonstrates the reliability and applicability of this approach in real-world breeding scenarios. Further research is highly encouraged using multispectral imagery technology and supervised deep learning methods to enhance classification accuracies. But as a cost-effective choice for breeders and associated researchers, UAV based RGB imagery system can be the solution for plant breeding programs in identifying the superior genotypes by identifying the lodged genotypes.

## Figures and Tables

**Figure 1 plants-12-02893-f001:**
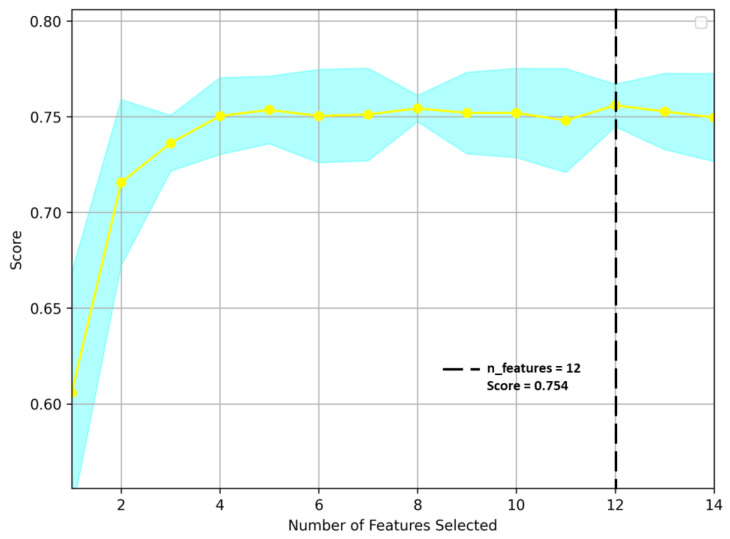
Recursive feature elimination (RFE) for image feature selection. Yellow line indicates the feature scores obtained during the RF-RFE process.

**Figure 2 plants-12-02893-f002:**
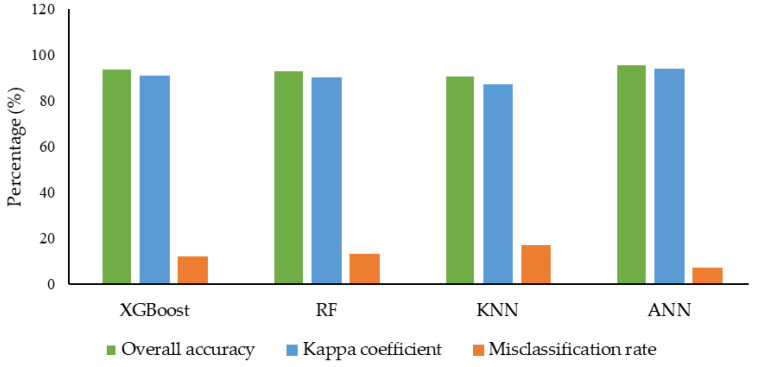
Overall accuracy, kappa coefficient and misclassification rate were achieved using SMOTE-ENN balanced dataset and four ML (XGBoost, RF, KNN, and ANN) classifiers.

**Figure 3 plants-12-02893-f003:**
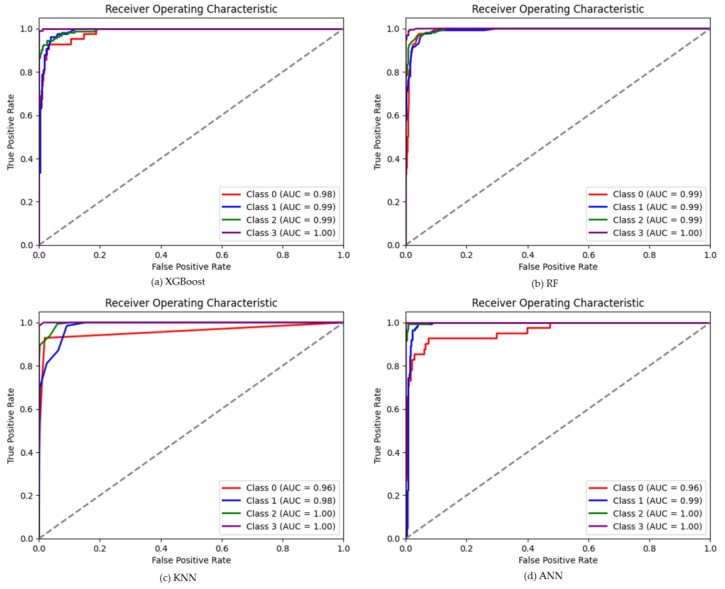
The ROC curve using SMOTE-ENN datasets and four ML classifiers, (**a**) for XGBoost (**b**) for RF (**c**) for KNN, and (**d**) for ANN classifier.

**Figure 4 plants-12-02893-f004:**
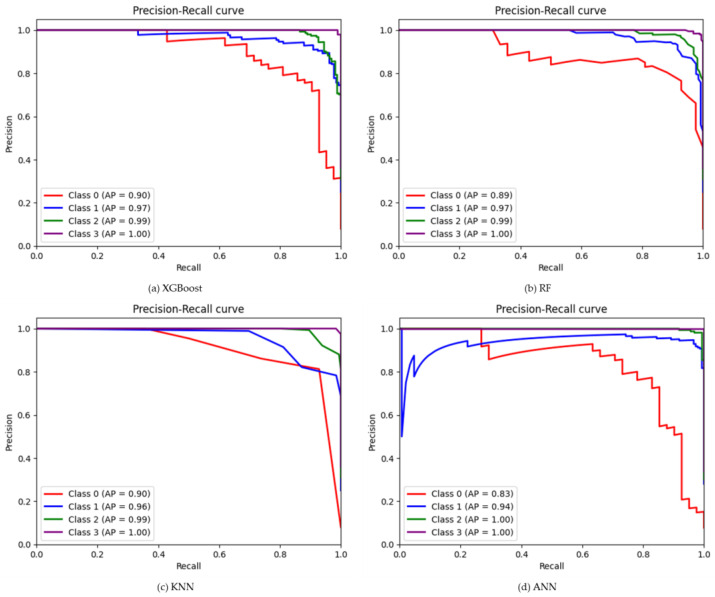
The Precision-Recall curve using SMOTE-ENN datasets and four ML classifiers, (**a**) for XGBoost (**b**) for RF (**c**) for KNN, and (**d**) for ANN classifier.

**Figure 5 plants-12-02893-f005:**
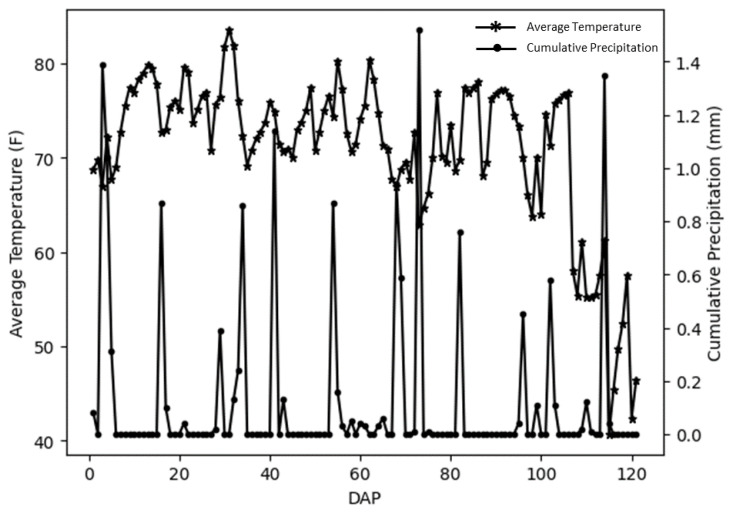
Daily average temperature and daily precipitation during the growing season in the experimental field. The asterisk and round marks are the day of data collection.

**Figure 6 plants-12-02893-f006:**
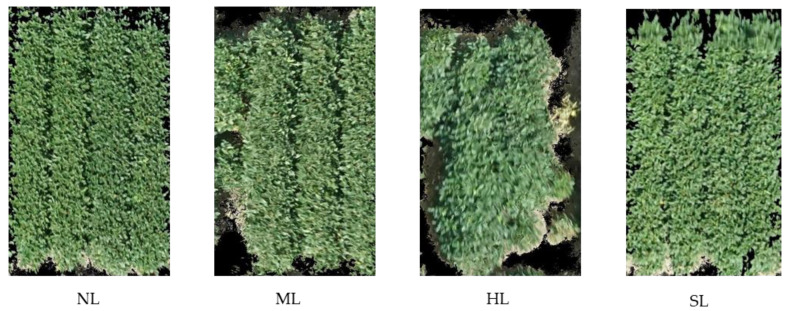
Ground-based classification scale of filed plots of soybean lodging with four classes, i.e., non-lodging (NL), moderate lodging (ML), high lodging (HL), and severe lodging (SL). Images were obtained at a height of 30 m from the ground using UAV.

**Figure 7 plants-12-02893-f007:**
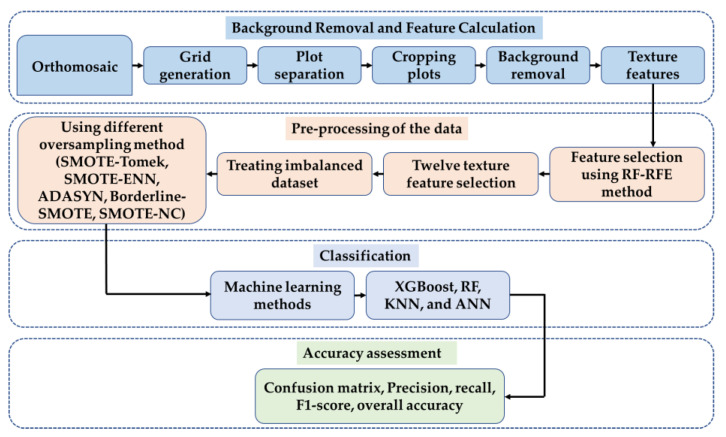
The flowchart of the soybean lodging classification.

**Table 1 plants-12-02893-t001:** Comparison of datasets before and after preprocessing for each classifier.

Dataset	NL	ML	HL	SL
Original	964	206	85	11
SMOTE-Tomek Link	923	964	964	964
SMOTE-ENN	224	700	784	925
Borderline-SMOTE	964	964	964	964
SMOTE-NC	964	964	964	964
ADASYN	964	976	966	962

**Table 2 plants-12-02893-t002:** Confusion matrix and model performance metrics of four lodging classes of soybean using machine learning classifiers (XGBoost, RF, KNN, and ANN).

Classifier	Actual Samples	NL	ML	HL	SL	Precision	Recall	F1-Score
XGBoost	NL	27	9	5	1	0.93	0.64	0.76
	ML	1	127	4	0	0.87	0.96	0.91
	HL	1	9	153	0	0.94	0.94	0.94
	SL	0	1	1	188	0.99	0.99	0.99
RF	NL	25	12	4	1	0.89	0.60	0.71
	ML	3	123	4	2	0.87	0.93	0.90
	HL	0	6	156	1	0.94	0.96	0.95
	SL	0	0	2	188	0.98	0.99	0.98
KNN	NL	23	17	1	1	0.92	0.55	0.69
	ML	1	115	12	4	0.82	0.87	0.84
	HL	1	9	153	0	0.92	0.94	0.93
	SL	0	0	0	190	0.97	1.00	0.99
ANN	NL	22	15	3	1	0.53	0.92	0.67
	ML	1	147	0	0	0.99	0.91	0.95
	HL	1	0	158	0	0.99	0.98	0.98
	SL	0	0	0	179	1.00	0.99	0.99

Where, soybean lodging classes are NL = Non-lodging, ML = Moderate lodging, HL = High lodging, and SL = Severe lodging. Data processing methods are SMOTE-Tomek Link = Synthetic Minority Oversampling Method with Tomek Link, SMOTE-ENN = Synthetic Minority Oversampling Method with Edited Nearest Neighbor, Borderline-SMOTE = Borderline method with Synthetic Minority Oversampling, SMOTE-NC = Synthetic Minority Oversampling Method with Nominal Continuous ADASYN = Adaptive Synthetic method and OA = overall accuracy.

**Table 3 plants-12-02893-t003:** Comparative analysis of crop classification results: soybean lodging classification in our study vs. previous research results on different crop lodging.

Crop/Land Cover	Phenotype/Dataset	Preprocessing	Analytical Method	Overall Accuracy	UAV Imagery Type	Reference
Other Researcher’s						
Maize	Lodging	SMOTE-ENN	XGBoost	94.5%	RGB	[47]
Wheat	Yield	Undersampling	RF	99.78%		[48]
Wheat	Yield	SMOTE	RF	99.92%		[48]
Three different land coverings	AVIRIS Indian Pines, Salinas Valley, Pavia University	SMOTE-Tomek Links	Rotation Forest (RoF) and Random Rotation Ensemble Forest (RREF)	93.89% and 93.17%	Hyperspectral	[49]
Grass	Lodging	None	SVM	79.1%	RGB	[50]
Maize	Lodging	None	MLC	86.61%	Multispectral	[12]
Barley	Lodging	None	RF	99.7%	Multispectral	[14]
Wheat	Lodging	None	SVM	92.6%	MS	[34]
Rice	lodging	None	ANN	97.8%	Hyperspectral	[51]
This study						
Soybean	Lodging	SMOTE-ENN	XGBoost	94%	RGB	This study
Soybean	Lodging	SMOTE-ENN	RF	93%	RGB	This study
Soybean	Lodging	SMOTE-ENN	KNN	91%	RGB	This study
Soybean	Lodging	SMOTE-ENN	ANN	96%	RGB	This study

**Table 4 plants-12-02893-t004:** Texture feature equations and uses by researchers.

Texture Features	Equation	Reference
Angular second moment	$\sum_{i}\sum_{j}p(i,j{)}^{2}$	Wheat, soybean, rice and maize classification [61]
Contrast	$\sum_{k=0}^{N_{g}-1}k^{2}p_{x-y}(k)$	Crop disease and different crop classification [61,62]
Correlation	$\frac{\sum_{i=1}^{N_{g}}\sum_{j=1}^{N_{g}}\left(ij\right)p\left(ij\right)-\mu_{x}\mu_{y}}{\sigma_{x}\sigma_{y}}$	Weed classification, crop disease and different crop classification [61,62,63]
Variance	$\sum_{i=1}^{N_{g}}\sum_{j=1}^{N_{g}}{\left(i-u\right)}^{2}p(i,j)$	Olive, potato, wheat, and sugar beet classification, weed classification, crop disease and different crop classification, and crop type mapping [61,62,63,64,65]
Inverse difference moment	$\sum_{i=1}^{N_{g}}\sum_{j=1}^{N_{g}}\frac{1}{1+{\left(i-j\right)}^{2}}p(i,j)$	Land cover classification [66]
Sum average	$\sum_{i=2}^{{2N}_{g}}{ip}_{x+y}(i)$	Poultry carcass identification [67]
Sum variance	$\sum_{i=2}^{{2N}_{g}}{\left(i-f_{s}\right)}^{2}p_{x+y}(i)$	Poultry carcass identification [67]
Sum entropy	$\sum_{i=2}^{{2N}_{g}}p_{x+y}(i)log(p_{x+y}\left(i\right))$	Crop discrimination [68]
Entropy	$-\sum_{i=1}^{N_{g}}\sum_{j=1}^{N_{g}}p\left(i,j\right)log(p\left(i,j\right))$	Crop classification using UAV multispectral imagery [69]
Difference variance	$Variance\;of\;p_{x-y}$	Poultry carcass identification [67]
Difference entropy	$-\sum_{i=0}^{N_{g}}p_{x-y}\left(i\right)log(p_{x-y}\left(i\right))$	Poultry carcass identification [67]
Information measures of correlation I	$\frac{Entropy-HXY1}{max(HX,HY)}$	Plant identification [70]
Information measures of correlation II	${[1-\exp\left(-2\left(HXY2-Entropy\right)\right)}^{1/2}$	Plant identification [70]
Maximal correlation coefficient	${\left(Second\;largest\;eigenvalue\;of\;Q\right)}^{1/2}$ where, $Q\left(i,j\right)=\sum_{k}\frac{p\left(i,k\right)p(j,k)}{p_{x}\left(i\right)p_{y}(k)}$	Crop classification [71]

Where, *N_g_* denotes the grayscale level, and *p(i,j)* is the normalized grayscale value at the position *i* and *j* within the kernel, and its sum is 1. *µ_x_*, *µ_y_*, *σ_x_* and *σ_y_* are the means and standard deviations of *p_x_* and *p_y_*. Also, *HX* and *HY* are the entropies of *p_x_* and *p_y_*.

## Data Availability

The data presented in this study are available on request from the corresponding author.

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
