# Peer review of "Assessment of Soybean Lodging Using UAV Imagery and Machine Learning"

_plants, 2023, doi:10.3390/plants12162893_

Round 1

Reviewer 1 Report

Comments and Suggestions for Authors

This paper considers the potential of unmanned aerial vehicle (UAV)-based imagery and machine learning in assessment of lodging conditions of soybean breeding lines.

Generally, lacks a good contextualization of the problem and a real scope. It would be appropriate to add some paragraphs concerning for example precision agriculture (e.g., proximal/remote sensing) and artificial intelligence.

I will consider the work for a publication only if a thorough review will be done.

Minor comments

Not all acronyms are spliced. Report either an initial table with all the explanations acronyms or always specify them before writing.

Generally, figure’s captions are not very detailed and explanatory.

Eliminate the “we” pronouns throughout the text.

Major comments

Introduction

This paragraph extensively describes the importance of soybean plant lodging, but no reference is made to the methodology used. In particular a section reporting the main application of the precision agriculture, such as for example, the proximal and/or remote sensing, could be added. In addition, since you are analyzing the images with the artificial neural networks, a section on artificial intelligence should also be included.

Describes and reports application examples concerning proximal sensing, image analysis, etc.

There is no real aim. Insert it to make the activity more understandable.

Conclusions

This section is not well organized. Please modified.

Author Response

We extend our appreciation to the reviewer for the diligent evaluation of our manuscript. Your thoughtful comments and suggestions are invaluable in strengthening the content and methodology of our research. We truly value the time and effort you dedicated to critically reviewing our work, and we are thankful for the constructive feedback that has helped us enhance the overall quality of our study. We have revised all the required comments in the new version of manuscript.

Minor comments

  1. Not all acronyms are spliced. Report either an initial table with all the explanations acronyms or always specify them before writing.

Response: Checked all acronyms thoroughly and all the acronyms are described prior to use.

  1. Generally, figure’s captions are not very detailed and explanatory.

Response: More information was included in the caption of Figures 2, 6 and 7.

  1. Eliminate the “we” pronouns throughout the text.

Response: Eliminated all “we” pronouns from the manuscript.

Introduction

  1. This paragraph extensively describes the importance of soybean plant lodging, but no reference is made to the methodology used. In particular a section reporting the main application of the precision agriculture, such as for example, the proximal and/or remote sensing, could be added. In addition, since you are analyzing the images with the artificial neural networks, a section on artificial intelligence should also be included.

Response: Added a particular section in the Introduction from line 66-78, describing the main application of the precision agriculture.

  1. Describes and reports application examples concerning proximal sensing, image analysis, etc.

Response: Same as the response to Response 4. 

  1. There is no real aim. Insert it to make the activity more understandable.

Response: We believe the aim and goal has been described in the original manuscript from the line 140-146.

Conclusions

  1. This section is not well organized. Please modified.

Response: Pertinent information has been added in lines 559-573, to make the conclusion clear.

Reviewer 2 Report

First of all, I want to congratulate the authors for their effort in this paper. They have presented the use of different pre-processing and classification methods to determine the soybean lodging using UAV RG images and ML algorithms. Their results show that proposal achieves high performance and there is a considerable dataset that supported their results. There are some minor aspect to impove before accepting the paper. Following, I include a list of comments aimed at enhancing the quality of their manuscript:

1.       The contextualisation of phenotyping using drone images for additional purposes than lodging must be included in the introduction. Fins some examples of alternative uses of phenotyping such as the identification of drought-resistant plants (10.1016/j.agwat.2022.107581) or pests-resistant plants (10.1016/j.biosystemseng.2016.04.010)

2.       The subsection "2.1. Field Experiment and Ground Data Collection 112 " should be divided into two subsections.

3.       In the first subsection, the authors have to describe the conditions of studied plots and in the second, the data collection. Regarding the field conditions, please indicate the distance between the plots and the meteorological station that provide data.

4.       Classification of lines 125-132 as based on the previous literature, or is a method defined for this experiment? Please clarify it.

5.       In Figure 2, please provide the scale or the height at which images were obtained in the caption.

6.       For equation (1), I suggest the authors use the most well-known formula for defining the accuracy (Accuracy = TP + TN TP + TN + FP + FN.)

7.       Indicate how the overall accuracy has been calculated.

8.       I suggest dividing the discussion section into different subsections and adding a comparative table in which the characteristics of experiments and the results are compared with current literature in terms of species, number of individuals, number of used ML algorithms, obtained metrics etc… to demonstrate the contributions of the paper in a straightforward way. The limitations (474-483) should be included as an independent subsection and linked with future work. Consider also adding a subsection detailing the potential impact of this research.

9.       In the conclusion section, divide the paragraph into two, keeping all the results and analysis in the second one.

Minor issues:

Avoid using, as a keyword, terms already used in the title. Deleted keywords included in the title and provided new keywords.

Author Response

We extend our appreciation to the reviewer for the diligent evaluation of our manuscript. Your thoughtful comments and suggestions are invaluable in strengthening the content and methodology of our research. We truly value the time and effort you dedicated to critically reviewing our work, and we are thankful for the constructive feedback that has helped us enhance the overall quality of our study. We have revised the manuscript accordingly based on your comments and suggestions.

  1. The contextualisation of phenotyping using drone images for additional purposes than lodging must be included in the introduction. Fins some examples of alternative uses of phenotyping such as the identification of drought-resistant plants (10.1016/j.agwat.2022.107581) or pests-resistant plants (10.1016/j.biosystemseng.2016.04.010)

Response: References have been added to the Introduction section to explain the concept of phenotyping. Pertinent information was added in Lines 83-92.

  1. The subsection "2.1. Field Experiment and Ground Data Collection 112 " should be divided into two subsections.

Response: The subsection "2.1. Field Experiment and Ground Data Collection’ is separated in the updated manuscript and highlighted.

  1. In the first subsection, the authors have to describe the conditions of studied plots and in the second, the data collection. Regarding the field conditions, please indicate the distance between the plots and the meteorological station that provide data.

Response: The studied field was a preliminary yield trial (PYT) field and were planted on June 3, 2019. Pertinent information was added to the subsection “2.1 Field Experiment” in lines 154-156, 159-161, and 162-165 to address the comments.

  1. Classification of lines 125-132 as based on the previous literature, or is a method defined for this experiment? Please clarify it.

Response: A line has been added to the subsection “2.2. Ground Data Collection” mentioning that the assigned four classes based on the visual scores were developed for this study only, after discussing with the experienced soybean breeders in line 179-181.

  1. In Figure 2, please provide the scale or the height at which images were obtained in the caption.

Response: Images were obtained using DJI Matrice 600 Pro at a height of 30 m. Pertinent information was added in Lines 189-190.

  1. For equation (1), I suggest the authors use the most well-known formula for defining the accuracy (Accuracy = TP + TN TP + TN + FP + FN.)

Response: The accuracy was not used in this study.

  1. Indicate how the overall accuracy has been calculated.

Response: The Overall accuracy (OA) was defined in Eq. 1.

  1. I suggest dividing the discussion section into different subsections and adding a comparative table in which the characteristics of experiments and the results are compared with current literature in terms of species, number of individuals, number of used ML algorithms, obtained metrics etc… to demonstrate the contributions of the paper in a straightforward way. The limitations (474-483) should be included as an independent subsection and linked with future work. Consider also adding a subsection detailing the potential impact of this research.

Response: A comparative table has been added to section “4. Discussion” in line 489-490, addressing Comparative analysis of crop classification results: soybean lodging classification in our study vs. previous research results on different crop lodging. The limitations have been included in the subsection “4.1. Future Work” (highlighted). The potential impact of the research has been added to conclusion from line 564-569.

  1. In the conclusion section, divide the paragraph into two, keeping all the results and analysis in the second one.

Response: The results were divided from the conclusion section .

  1. Avoid using, as a keyword, terms already used in the title. Deleted keywords included in the title and provided new keywords.

Response: Accepted. Keywords have been updated in line 35.

Round 2

Reviewer 1 Report

The manuscript in the present form can be published in Plants as the authors have been improved it.

I don’t need to review another version because I accept the work in the present form.

Best regards

Reviewer 2 Report

The authors have correctly addressed all the comments and the paper is now ready to be acepted.